# Hardware-in-the-Loop Simulation of Self-Driving Electric Vehicles by Dynamic Path Planning and Model Predictive Control

**Yi Chung [1] and Yee-Pien Yang [1,2,\*]**

[1]  Department of Mechanical Engineering, National Taiwan University, Taipei 10617, Taiwan;
    ms0227162@gmail.com
[2]  Mechanical and Mechatronics Systems Research Laboratories, Industrial Technology Research Institute (ITRI),
    Hsinchu 31057, Taiwan
\*  Correspondence: ypyang@ntu.edu.tw or yeeyang@itri.org.tw; Tel.: +886-2-3366-2682

**Abstract:** This paper applies a dynamic path planning and model predictive control (MPC) to simulate self-driving and parking for an electric van on a hardware-in-the-loop (HiL) platform. The hardware platform is a simulator which consists of an electric power steering system, accelerator and brake pedals, and an Nvidia drive PX2 with a robot operating system (ROS). The vehicle dynamics model, sensors, controller, and test field map are virtually built with the PreScan simulation platform. Both manual and autonomous driving modes can be simulated, and a graphic user interface allows a test driver to select a target parking space on a display screen. Three scenarios are demonstrated: forward parking, reverse parking, and obstacle avoidance. When the vehicle perceives an obstacle, the map is updated and the route is adaptively planned. The effectiveness of the proposed MPC is verified in experiments and proved to be superior to a traditional proportional–integral–derivative controller with regards to safety, energy-saving, comfort, and agility.

**Keywords:** electric van; self-driving vehicle; path planning; model predictive control

## 1. Introduction

The advanced development of science and technology has made many researchers put their efforts to enhance safety, energy-saving, comfort, and agility in road traffic through vehicle automation. Since the last two decades, many researchers have addressed how the raw images from environment and the steering control states of a vehicle could be used to drive a vehicle autonomously in real time [1]. Nearly 30 years have passed, and we still have not seen the commercialization of autonomous vehicles without a steering wheel. Various advanced driver assistance system (ADAS) modules have been put on the market, such as adaptive cruise control (ACC), autonomous emergency braking system (AEB), lane keeping system (LKS), etc. Although these features can help drivers reduce the burden of driving and improve road safety, they may still not able to solve congestion and parking problems in urban areas.

Recently, research has also focused on parking assist technologies, such as vision-based parking assist systems [2,3], ultrasonic-based auto parking systems [4], laser scanner radar-based parking systems [5], and parking assist systems using hybrid method [6]. While these research efforts ended with the function of assisted parking, more advanced challenges remain, for instance, to build a system that includes path planning and fully automated parking. First, the path planning must convey to the vehicle the knowledge of its surrounding environment so that the vehicle control unit can command its actuators with appropriate actions in real time. A variety of methods of path planning were based on the requirements of the problem to be solved, such as rapidly-exploring random trees (RRT) [7], heuristically-guided RRT [8], potential field algorithm [9], and A-star [10]. By adding kinematic constraints to the A-star, a hybrid A-star algorithm was proposed by

Dolgov et al. [11], which had been used by the Stanford team to execute U-turns on blocked roads and to navigate vehicles in parking lots in a grand challenge project supported by the Defense Advanced Research Projects Agency (DARPA).

After the path planning task sends signals to command a vehicle to track the expected path, the vehicle control unit will drive the vehicle by considering its coupled longitudinal and lateral dynamic behaviors. For a small low-speed harvester-assisting vehicle, Loukatos [12] simply used its geometrical model to calculate speed controls to drive motors to reach a specific turning angle. For high-speed normal vehicles with nonlinear and/or time-varying dynamics, traditional PID controller could scarcely be used to maneuver such vehicles of non-holonomic motions. Alternative methods of gain scheduling, parameter adaptation, and auto-tuning might be able to upgrade PID control performances [13] with or without vehicle models. More advanced techniques, such as sliding mode control [14], pure pursuit control [15,16], optimal predictive control [17], iterative linear quadratic regulator (LQR) [18], nonlinear model predictive control [19], and robust control [20] have been proposed to control the vehicle with complex dynamic behaviors.

One of the most promising methods of autonomous vehicle control is model predictive control (MPC), which accommodates both kinematic and dynamic vehicle models to improve the vehicle tracking performance at low and high speeds [21]. Either nonlinear MPC or time-varying linear MPC is attractive to the application of trajectory tracking for self-driving vehicles, and has been proved stable in the sense of Lyapunov theorem [22]. Vougioukas [23] applied a nonlinear model predictive tracking controller to a mobile robot in the presence of obstacles. Only its kinematic model was used to compute an optimal M-step-ahead control sequence in real time, by minimizing the cost of tracking errors and control efforts. However, the computational load may impede fast applications to self-driving vehicles [24].

This paper combines hybrid A-star path planning and time varying linear MPC with vehicle dynamic models for an electric van (e-van) to simulate self-driving and parking on a hardware-in-the-loop (HiL) simulation platform. The HiL simulation technique is often used for testing the proposed path planning and model predictive control strategy before it is implemented on a real vehicle in the real-world environment. In this paper, the hardware part has power steering wheel and motor, accelerator and brake pedals and their control unit, and computers. Other vehicle physical parts, components, subsystems, and vehicle dynamics are replaced by mathematical models coded in the software platform on computers. Since simulations are performed on a virtual plant, no physical plant could be damaged, and no people could be injured.

Section 2 establishes the kinematic and dynamic vehicle models. Section 3 briefs the hybrid A-star path planning algorithm. Section 4 presents the MPC by minimizing a cost function that describes vehicle performance in terms of safety, energy-saving, comfort, and agility. Section 5 introduces experimental results with the HiL platform and Section 6 concludes the research.

## 2. Vehicle Model

The e-van discussed in this paper is a Chung-Hua Motor Company commercial vehicle with a gross weight of 1.46 tons and 2 seats, which is driven by a 45 kW (rated) interior permanent magnet motor (IPM) and a 32-kWh lithium battery bank.

### 2.1. Kinematic Model

In Figure 1, the bicycle model of the e-van is introduced with a mass center at $C$, from which $L_f$ and $L_r$ are, respectively, the distances to the front and rear tires. With respect to the instantaneous center of zero velocity at $O$, the slip angles of the front and rear tires are denoted by $\alpha_f$ and $\alpha_r$, the yaw angle is $\psi$, the side slip angle of the vehicle is $\beta$, and the steering angle is $\delta$.

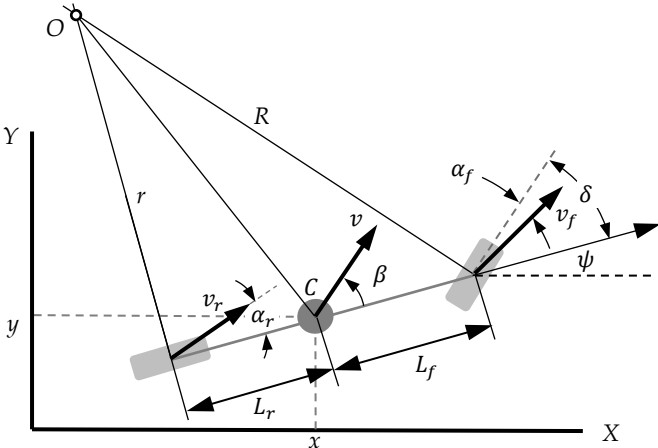

**Figure 1.** Bicycle model of the vehicle.

The velocities $V_f$ and $V_r$ of the front and rear wheels are derived as

$$V_f = \begin{bmatrix} v\cos\beta \\ v\sin\beta + L_f\dot\psi \\ 0 \end{bmatrix} = \begin{bmatrix} v_f\cos(\delta - \alpha_f) \\ v_f\sin\left(\delta - \alpha_f\right) \\ 0 \end{bmatrix} \tag{1}$$

and

$$V_r = \begin{bmatrix} v\cos\beta \\ v\sin\beta - L_r\dot\psi \\ 0 \end{bmatrix} = \begin{bmatrix} v_r\cos\alpha_r \\ -v_r\sin\alpha_r \\ 0 \end{bmatrix}, \tag{2}$$

where $v$ is the vehicle speed. When both the side slip angle and steering angle are small, i.e., $\tan(\delta - \alpha_f) \approx \delta - \alpha_f$, $\beta \approx 0$, and $\tan\alpha_r \approx \alpha_r$, their relationships are simply described as

$$\alpha_f = \delta - \beta - L_f\frac{\dot\psi}{v} \text{ and } \alpha_r = -\beta + L_r\frac{\dot\psi}{v}. \tag{3}$$

Therefore, the lateral force on the front tire and the lateral force on the rear tire are derived by the Magic Formula tire model, as follows

$$F_{yf} = C_{\alpha_f}\alpha_f \text{ and } F_{yr} = C_{\alpha_r}\alpha_r, \tag{4}$$

where $C_{\alpha f}$ is the cornering stiffness of the front tire and $C_{\alpha r}$ is the cornering stiffness of the rear tire. When the tire side slip angles $\alpha_f$ and $\alpha_r$ and the vehicle side slip angle $\beta$ are small, the kinematic equation can be derived as

$$\begin{bmatrix} \dot{x} \\ \dot{y} \\ \dot{\psi} \end{bmatrix} = \begin{bmatrix} \cos\psi \\ \sin\psi \\ \tan\frac{\delta}{L} \end{bmatrix} v_r = \begin{bmatrix} \cos\psi \\ \sin\psi \\ 0 \end{bmatrix} v + \begin{bmatrix} 0 \\ 0 \\ 1 \end{bmatrix}\dot\psi, \tag{5}$$

where $\dot{x} = \dot{x}_r$ and $\dot{y} = \dot{y}_r$ are the velocity components at the center of rear wheel. Since the rear tire side slip angle is negligibly small, the velocity of rear axle is equal to the velocity of mass center, i.e., $v_r = v$.

*2.2. Dynamic Model*

As shown in Figure 2, the longitudinal traction forces are usually simplified as $F_{xr} = F_{xr1} = F_{xr2}$ and $F_{xf} = F_{xf1} = F_{xf2}$ when the vehicle moves along a straight line. The force with subscript 1 stands for the force exerted on the left tire, and subscript 2 stands for

the force exerted on the right tire. The following equations represent the normal forces $N_f$ and $N_r$ of the front and rear wheels and the tractive force $F_x$:

$$N_f = \left( mgL_r \cos\theta - \frac{1}{2}\rho C_d A_f v_x^2 h_a - mh_m \dot{v}_x \right) / 2L, \tag{6}$$

$$N_r = \left( mgL_f \cos\theta + \frac{1}{2}\rho C_d A_f v_x^2 h_a + mh_m \dot{v}_x \right) / 2L, \tag{7}$$

$$F_x = 2\left( F_{xf} + F_{xr} \right) = m\dot{v}_x + mg\sin\theta + \frac{1}{2}\rho C_d A_f v_x^2 + 2C_t\left( N_f + N_r \right). \tag{8}$$

Here, the geometric parameters are: $L = L_f + L_r$, $A_f$ the frontal area of vehicle, $h_a$ the equivalent height of aerodynamic point, $h_m$ the height of mass center, and $\theta$ the slope angle in degrees. The kinematic and kinetic parameters are: $m$ the vehicle mass, $g$ the gravity acceleration, $\rho$ the air density, $v_x$ the longitudinal vehicle velocity, $C_d$ the aerodynamic coefficient, $C_t$ the friction coefficient between tire and ground, $F_{xf}$ the traction force on the front tire, and $F_{xr}$ the traction force on the rear tire.

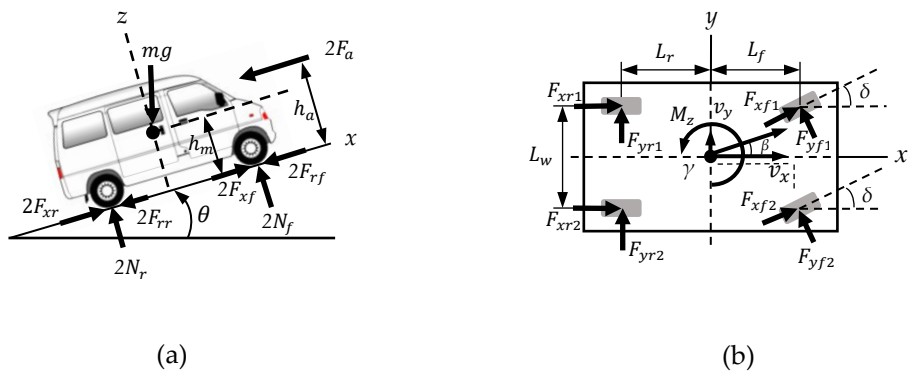

(a)            (b)

**Figure 2.** Vehicle dynamics models in the (**a**) longitudinal and (**b**) lateral directions.

When the vehicle is cornering, the four-wheel model is used to describe the vehicle dynamics equations in the lateral, longitudinal, and yaw directions, as follows:

$$(F_{xf1} + F_{xf2})\cos\delta - (F_{yf1} + F_{yf2})\sin\delta + F_{xr1} + F_{xr2} = m\left[ \dot{v}_x - \left( \gamma + \frac{d\beta}{dt} \right) v_y \right], \tag{9}$$

$$(F_{xf1} + F_{xf2})\sin\delta + (F_{yf1} + F_{yf2})\cos\delta + F_{yr1} + F_{yr2} = m\left[ \dot{v}_y + \left( \gamma + \frac{d\beta}{dt} \right) v_x \right], \tag{10}$$

$$L_f(F_{yf1} + F_{yf2})\cos\delta - L_r(F_{yr1} + F_{yr2}) + \frac{L_w}{2}\left( F_{yf1} - F_{yf2} \right)\sin\delta + M_z = I_z \frac{d}{dt}\left( \gamma + \frac{d\beta}{dt} \right), \tag{11}$$

$$M_z = L_f(F_{xf1} + F_{xf2})\sin\delta + \frac{L_w}{2}(-F_{xf1} + F_{xf2})\cos\delta + \frac{L_w}{2}(-F_{xr1} + F_{xr2}), \tag{12}$$

where $F_{xf1}$ is the longitudinal traction force on the left front tire and $F_{xf2}$ is the longitudinal traction force on the right front tire. They are assumed equal when the steering angle $\delta$ is zero. The lateral traction force on the left front tire is denoted by $F_{yf1}$ and the lateral traction force on the right front tire is denoted by $F_{yf2}$. Similarly, $F_{xr1}$ is the longitudinal traction force on the left rear tire, $F_{xr2}$ is the longitudinal traction force on the right rear tire, $F_{yr1}$ is the longitudinal traction force on the left rear tire, and $F_{yr2}$ is the longitudinal traction force on the right rear tire. In the yaw direction, $I_z$ represents the mass moment of inertia, $M_z$ is the yaw moment for cornering, and $\gamma = \dot{\psi}$ is the yaw velocity.

Equations (9)–(12) are used to calculate torque distributions as the vehicle is cornering at low speeds. Then, the pitch and roll motions can be omitted, and the rolling resistance, aerodynamic drag, and hill climbing resistance of the last terms in (6) can be neglected for simplicity.

## 3. Hybrid A-Star Algorithm

A crucial part for a self-driving vehicle is the path planning system, which typically encompasses different abstract layers, such as mission and motion planning. Mission planning consists of trajectory planning and path planning. The path planning generates a collision free path in an obstacle environment by optimizing it with regard to some rules, while the trajectory planning schedules the vehicle movement on the planned path. The motion planner, on the other hand, operates at a lower level to avoid obstacles while progressing towards local goals.

This paper applies the hybrid A-star algorithm [11,25] of path planning for an auto parking system. The first phase of this approach is different from the well-known A-star algorithm, which uses 3D kinematic states of the vehicle, but we modify the state-update rule to obtain the continuous-state data on the discrete searching grids on a map. The traditional A-star path planning only allows reaching centers of cells connected with piecewise linear lines. However, the hybrid A-star allows expansion towards any continuous point on the grid as a state, as shown in Figure 3. The state is described by x = $(x, y, \psi)$, in which $x$ and $y$ denote the position of the mass center $C$ and $\psi$ the heading of the vertex or the yaw angle of the vehicle in Figure 1.

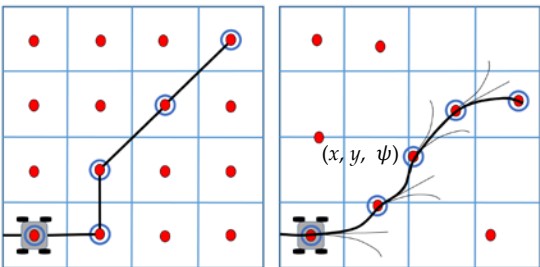

**Figure 3.** Traditional A* searching path (**left**) vs. hybrid A* searching path (**right**).

The non-holonomic constraints of the vehicle are considered for the vertex expansion by taking one of three actions—maximum steering left, maximum steering right, or no steering. In Figure 4, the vertex search starts from the current vehicle state $x_s$ to the goal state $x_g$ which are provided by the PreScan simulation platform. Then, the hybrid A-star will generate six successor vertices, three of which are driving forward, while three are driving in reverse (not shown in the figure). Each vertex expansion is generated by the minimal turning radius $r = L/\tan\delta$ based on the vehicle parameters in Figure 1 to make sure that the resulting paths are always drivable. When a new vertex is expanded to a cell already occupied by other vertices, the ones with the higher cost will be deleted and the vertex with the least cost remains.

Any point $x_n$ on the planned path satisfies the optimal value of the cost function

$$f(x_n) = g(x_n) + h(x_n),\tag{13}$$

where $g(x_n)$ is the distance or cost between $x_s$ and $x_n$, $h(x_n)$ is the predicted distance or cost from $x_n$ to $x_g$, namely, the heuristic function.

The heuristic function used in the hybrid A-star path planning consists of a constrained heuristic and an unconstrained heuristic. The constrained heuristic ignores environmental obstacles but incorporates the geometric and dynamic constraints of the vehicle. The unconstrained heuristic disregards vehicle constraints but only accounts for obstacles.

In this paper, the Reeds-Shepp curve [26] was used as the constrained heuristic function for searching a path, which referred to the shortest curve that considered the heading

and curvature restrictions. Taking the curvature limit of our vehicle into account, the Reeds-Shepp curve could generate drivable way points for the vehicle. As to the unconstrained heuristic function, the traditional A-star with Euclidean distance was introduced. By combining both constraint and unconstraint heuristic functions, the final path was generated to satisfy geometry constraints as well as to avoid obstacles.

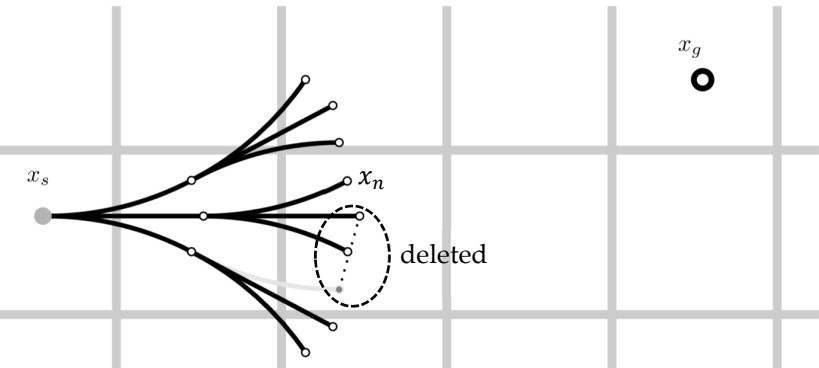

**Figure 4.** Forward part in the path planning process on PreScan platform.

In experiments, the size of each grid cell was $0.5 \times 0.5$ m$^2$ to compose the scene of the test field on the PreScan simulation platform. As the vehicle was entering the parking lot of a size about $65 \times 45$ m$^2$, it was switched to the self-driving mode. The Nvidia AutoChauffeur PX2, which had a Cortex A57 microarchitecture, took about 0.5–1.5 s to plan the path from the entrance of the parking lot to the selected parking space. During the movement, the vehicle speed was under 10 km/h; the control actions were determined in a sampling period of 100 ms after collecting the real-time information of sensors and waypoints along the path.

Though a vehicle model used to expand vertices might result in an excessive steering action, it can be easily solved by the proposed MPC optimization. Figure 5 illustrates a visualization process of path planning on the parking lot of testing site created in the graphical user interface (GUI) of the ROS visualization tool (RViz).

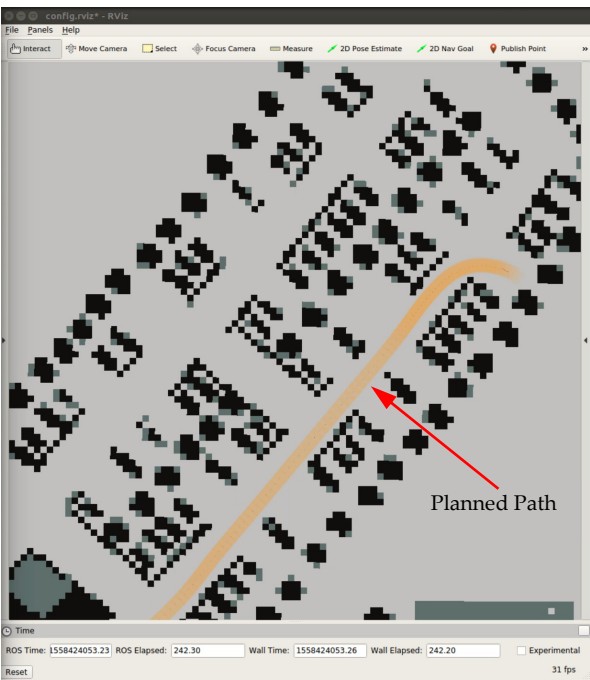

**Figure 5.** A visualization process of path planning on the parking lot created by RViz.

## 4. Model Predictive Control

The MPC that imitates the working pattern of the human brain is here applied to autonomous vehicles under various constraints. Figure 6 illustrates the conceptual diagram of MPC. The measured or estimated system states are sent to the MPC block, where the optimal control action is calculated by a system model and constraints to track the pre-filtered set points by minimizing a cost function.

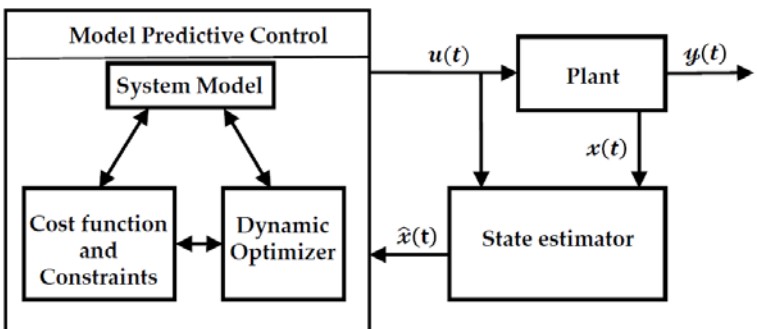

**Figure 6.** Structure of model predictive control.

### 4.1. Vehicle States Prediction Model

The system model in Figure 6 is responsible for vehicle states prediction. The kinematic model (5) can be linearized about a reference state $\overline{x} = [x_r, y_r, \psi_r]^T$ and a reference control input $\overline{u} = [v_r, \delta_r]^T$ by Taylor series expansion after neglecting high-order terms:

$$\dot{\hat{x}}(t) = A_1 \hat{x}(t) + B_1 u_1(t), \tag{14}$$

$$A_1 = \begin{bmatrix} 0 & 0 & -v_r \sin \psi_r \\ 0 & 0 & v_r \cos \psi_r \\ 0 & 0 & 1 \end{bmatrix}, \tag{15}$$

$$B_1 = \begin{bmatrix} \cos \psi_r & 0 \\ \sin \psi_r & 0 \\ \frac{\tan(\delta)}{L} & \frac{v_r}{L \cos^2 \delta_r} \end{bmatrix}, \tag{16}$$

where $\hat{x} = [x - x_r, y - y_r, \psi - \psi_r]^T$, $u_1 = [v - v_r, \delta - \delta_r]^T$.

Under the assumptions of a small steering angle, $\sin \delta \approx 0$, and $\cos \delta \approx 1$, and small changing rate of side slip angle, $d\beta/dt = 0$, dynamic Equations (6)–(12) can be simplified with the Magic Formula tire model (4) as

$$m\dot{v}_y = -mv_x\psi + 2\left[C_{\alpha f}\left(\delta - \frac{v_y + L_f\dot{\psi}}{v_x}\right) + C_{\alpha r}\left(\frac{L_r\dot{\psi} - v_y}{v_x}\right)\right], \tag{17}$$

$$m\dot{v}_x = mv_y\psi + 2\left[C_t N_f + C_{\alpha f}\left(\delta - \frac{v_y + L_f\dot{\psi}}{v_x}\right) + C_t N_r\right], \tag{18}$$

$$I_Z\ddot{\psi} = 2\left[L_f C_{\alpha f}\left(\delta - \frac{v_y + L_f\dot{\psi}}{v_x}\right) - L_r C_{\alpha r}\left(\frac{L_r\dot{\psi} - v_y}{v_x}\right)\right], \tag{19}$$

$$\dot{X} = v_x \cos \psi - v_y \sin \psi, \tag{20}$$

$$\dot{Y} = v_x \sin \psi + v_y \cos \psi, \tag{21}$$

where $2C_t(N_f + N_r)$ represents the driving force or braking force exerted between tires and ground. Define a new state $z = [v_y, v_x, \psi, \dot{\psi}, Y, X]^T$, and input $u_2 = [\delta, p]$, where

$p = 2C_t(N_f + N_r)/m$ denotes the pedal angle of accelerator or brake. The linearized state-space model of (17)–(21) is derived as

$$\dot{z} = A_2(t)z(t) + B_2(t)u_2(t), \tag{22}$$

where

$$A_2(t) = \begin{bmatrix} \dfrac{-2(C_{af} + C_{ar})}{mv_x} & A_{12} & 0 & -v_x + \dfrac{2(L_rC_{ar} - L_fC_{af})}{mv_x} & 0 & 0 \\[2mm] \dot{\psi} - \dfrac{2C_{af}}{mv_x} & A_{22} & 0 & v_y - \dfrac{2L_fC_{af}}{mv_x} & 0 & 0 \\[2mm] 0 & 0 & 0 & 1 & 0 & 0 \\[2mm] \dfrac{2(L_rC_{ar} - L_fC_{af})}{I_z v_x} & A_{42} & 0 & \dfrac{-2}{I_z v_x}\left(L_f^2 C_{af} + L_r^2 C_{ar}\right) & 0 & 0 \\[2mm] \cos\psi & \sin\psi & A_{53} & 0 & 0 & 0 \\[2mm] -\sin\psi & \cos\psi & A_{63} & 0 & 0 & 0 \end{bmatrix}, \tag{23}$$

$$A_{12} = \frac{2}{mv_x^2}\left[C_{af}\left(v_y + L_f\dot{\psi}\right) + C_{ar}\left(v_y - L_r\dot{\psi}\right)\right] - \dot{\psi}, \tag{24}$$

$$A_{22} = \frac{2C_{af}}{mv_x^2}\left(v_y + L_f\dot{\psi}\right), \tag{25}$$

$$A_{42} = \frac{2}{I_z v_x^2}\left[L_f C_{af}\left(v_y + L_f\dot{\psi}\right) - L_r C_{ar}\left(v_y - L_r\dot{\psi}\right)\right], \tag{26}$$

$$A_{53} = v_x\cos\psi - v_y\sin\psi, \tag{27}$$

$$A_{63} = -v_y\sin\psi - v_x\cos\psi, \tag{28}$$

$$B_2(t) = \begin{bmatrix} \dfrac{2C_{af}}{m} & \dfrac{2C_{af}}{m} & 0 & \dfrac{2L_fC_{af}}{I_z} & 0 & 0 \\[2mm] 0 & 1 & 0 & 0 & 0 & 0 \end{bmatrix}^T. \tag{29}$$

### 4.2. Cost Function and Constraints

The cost function for the MPC is defined in a quadratic form:

$$J = w_{et}(e_t)^2 + w_{e\psi}(e_\psi)^2 + w_p(p)^2 + w_{\dot{p}}(\dot{p})^2 + w_\delta(\delta)^2 + w_{\Delta\delta}(\Delta\delta)^2 + w_{v\delta}(v\delta)^2 + w_v(v - v_r)^2, \tag{30}$$

where $e_t = \sqrt{(x - x_r)^2 + (y - y_r)^2}$ is the cross-tracking error from reference path, $e_\psi = \psi - \psi_r$ is the heading error from reference path, as shown in Figure 7. The weighting factors $w$'s account for the importance of vehicle safety, energy-saving, comfort, and agility. For safety and agility, $w_{et}$ weighs the cross-tracking error from reference path, and $w_{e\psi}$ the heading error from reference path. For energy-saving, $w_p$ weighs the pedal angle of accelerator or brake, and $w_{\dot{p}}$ the change of the pedal angle of accelerator or brake. For comfort and safety, $w_\delta$ weighs the steering angle, $w_{\Delta\delta}$ the change of steering angle, and $w_{v\delta}$ the coupling effect of vehicle speed and steering angle. For agility, $w_v$ weighs the vehicle speed deviation from the reference.

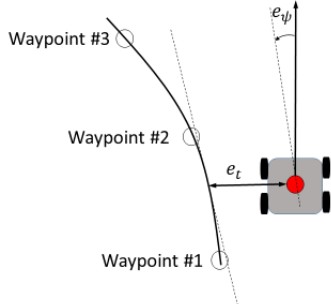

**Figure 7.** Definition of cross tracking and heading errors.

It is interesting to devise a rule of choosing reasonable values for the eight weighting factors. First, the eight weighting factors can be grouped into two categories: the weightings $w_{et}$, $w_{e\varphi}$, $w_v$, and $w_{v\delta}$ on the importance of vehicle states, and the weightings $w_\delta$, $w_{\Delta\delta}$, $w_p$, and $w_{\dot{p}}$ on the importance of control variables. The former category relates more about vehicle safety by evaluating the trajectory tracking and heading errors as well as speed errors, while the latter relates more about energy-saving and somewhat of driving comfort. Agility has something to do with fast response. In this paper, when we say that the self-driving vehicle is agile, it means that the vehicle returns quickly back to the planned reference trajectory from which the vehicle may deviate. Therefore, the weightings on $w_{et}$, $w_{e\varphi}$ and $w_v$ care both tracking errors and fast response to diminish the errors.

Safe driving is always the first rule for a self-driving vehicle. It is suggested in this paper that, in any cases, 60% or more weightings should be assigned to the vehicle states category which emphasizes safe driving, but less weightings be put to the control variables category to consider moderately the energy saving and comfort of driving.

The inequality constraints of control input $u$ and $\Delta u$ are hard constraints, while the predicted outputs are usually soft constraints, as follows:

$$u_{min}(k) \leq u(k+j) \leq u_{max}(k) y_{min}(k) \leq y(k+j) \leq y_{max}(k). \tag{31}$$

## 5. Experiments and Results

The HiL simulation platform for experiments encompasses two parts as shown in Figure 8. The hardware part in quadrants 1 and 2 consists of an Nvidia AutoChauffeur PX2, a power steering wheel with motor, and an accelerator and brake pedal and its controller (Logitech G29). The software part includes:

(1)　ROS on the Nvidia PX2 in quadrant 1, where the proposed hybrid A-star path planning algorithm and MPC algorithm are executed;

(2)　Vehicle kinematic and dynamic models model coded in the Matlab/Simulink platform in quadrant 3;

(3)　Sensor models of GPS, camera, radar, and lidar, and the world map for simulation provided by PreScan platform in quadrant 4.

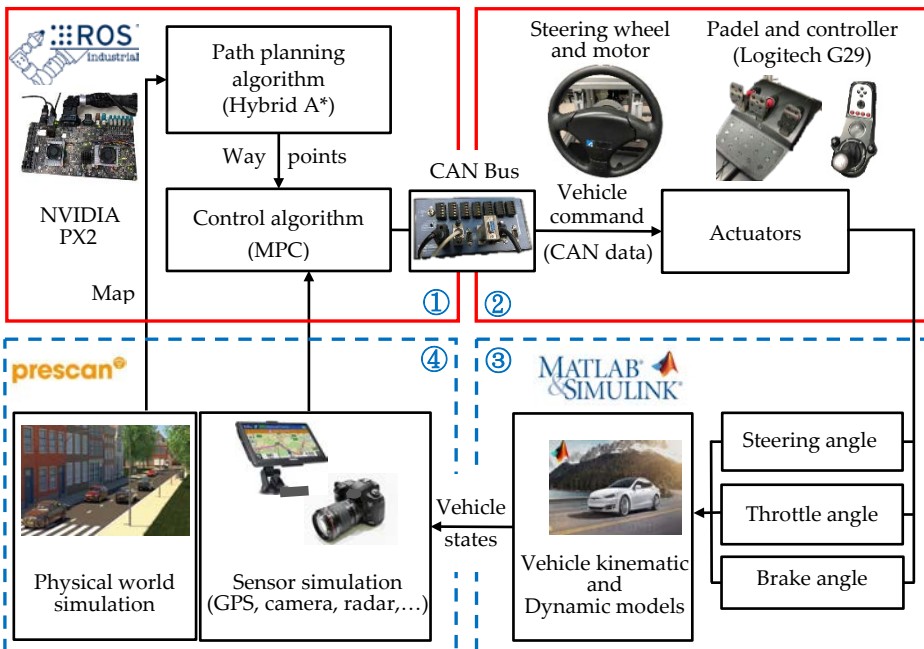

**Figure 8.** Architecture of HiL simulation platform and experimental rig.

The test field was on the ITRI campus, whose map was planned off-line by the hybrid A-star algorithm in the PreScan simulation platform. In each sampling period of 100 ms, the proposed MPC took from the map of planned path 20–40 waypoints ahead of the current position to calculate a new control action by applying an open source Interio Point Optimizer (Ipopt) [27]. The experimental setup is illustrated in Figure 9, and parts of the test field are illustrated in Figure 10. This section verifies the feasibility of the hybrid A-star path planning and the proposed MPC strategy through three different simulation scenarios.

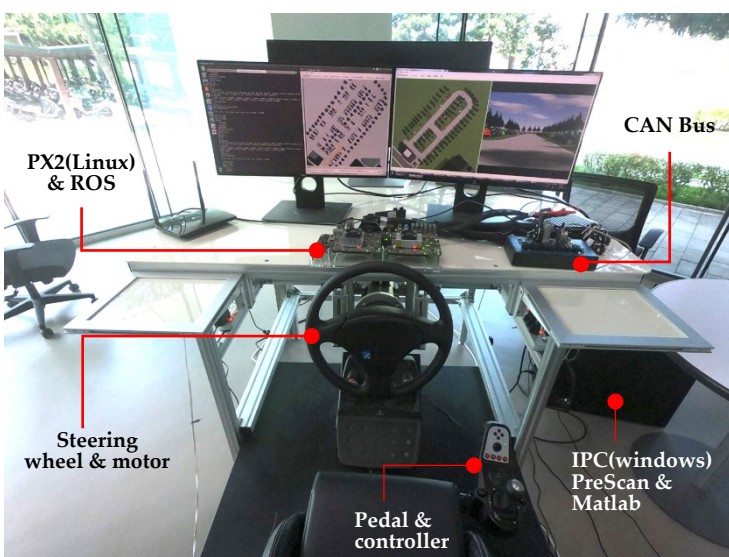

**Figure 9.** HiL experimental setup.

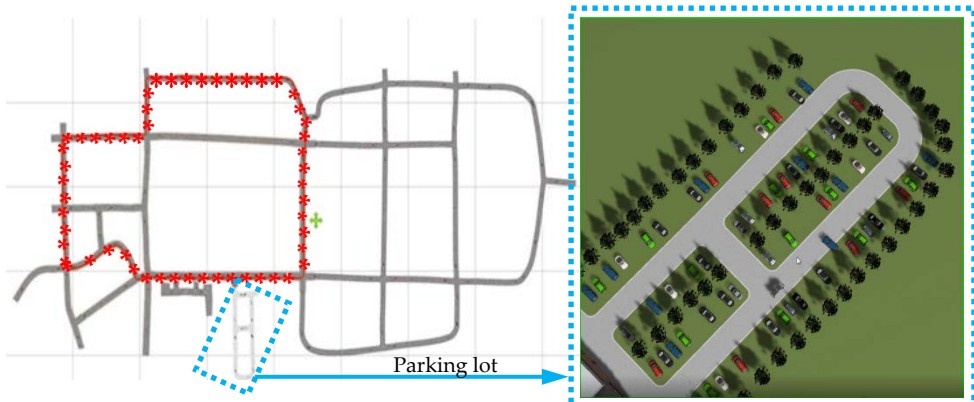

**Figure 10.** Self-driving route (red star line) by PreScan and parking lot map on the test field.

### 5.1. Self-driving Performance

In order to verify the independence and intuitiveness of the weighting factors in the cost function of the MPC strategy, two sets of weighting factors are assigned, as shown in Table 1. As was suggested, that in any cases 60% or more weighting should be assigned to the vehicle states category that emphasized safe driving. Setting 1 has 85% weighting and Setting 2 has 62.5% weighting on the vehicle states category. For more insight into the weighting factors on the tracking and heading errors, Setting 1 puts 27.5% each on $w_{et}$ and $w_{e\psi}$, but Setting 2 puts only 17.5% each on $w_{et}$ and $w_{e\psi}$. In other words, Setting 1 puts more importance on safety than Setting 2, in terms of 55% versus 35% on the weighting factors $w_{et}$ and $w_{e\psi}$.

**Table 1.** Weighting factors for self-driving test.

| Weightings | | Case 1 | Case 2 | Index of Importance * | | | |
|---|---|---|---|---|---|---|---|
| Unit | | % | % | S | E | C | A |
| Vehicle states | $w_{et}$ | 27.5 | 17.5 | ● | | | ▲ |
| | $w_{e\varphi}$ | 27.5 | 17.5 | ● | | | ▲ |
| | $w_v$ | 20 | 12.5 | ▲ | | | ● |
| | $w_{v\delta}$ | 10 | 15 | ▲ | | ● | |
| Control variables | $w_\delta$ | 1.5 | 7.5 | ▲ | | ▲ | |
| | $w_{\Delta\delta}$ | 3.5 | 10 | ▲ | | ▲ | |
| | $w_p$ | 4.5 | 5 | | | ● | |
| | $w_{\dot{p}}$ | 5.5 | 15 | | | ● | |

* **S** (safety), **E** (energy-saving), **C** (comfort), **A** (agility). ● means highly while ▲ partially correlated to S, E, C, or A.

In contrast, less importance is suggested for the category of control variables, considering the energy saving and comfort of the vehicle. For energy saving, Setting 2 assigns 20% but Setting 1 assigns only 10% weighting to $w_p$ and $w_{\dot{p}}$, which means that more importance on energy saving is given in Setting 2 than in Setting 1. For the importance of comfort, Setting 2 allocates 32.5% while Setting 1 allocates only 15% weighting to $w_\delta$, $w_{\Delta\delta}$, and $w_{v\delta}$, which means that Setting 2 may cause more driving comfort than Setting 1.

Figure 11, marked with driving ordering numbers, shows that on the prescribed route in Figure 10, Setting 1 has better tracking and heading performances than Setting 2, where more erroneous offsets are observed, especially at corners, from the prescribed route. Figure 12 also shows that the self-driving vehicle with Setting 1 ($w_v$ at 20%) follows the prescribed tracking speed at 10 km/h better than that of Setting 2 ($w_v$ at 12.5%). However, Figure 13 shows that the lateral acceleration curve of Setting 2 is much smoother and therefore more comfortable for passengers in the vehicle, than that of Setting 1.

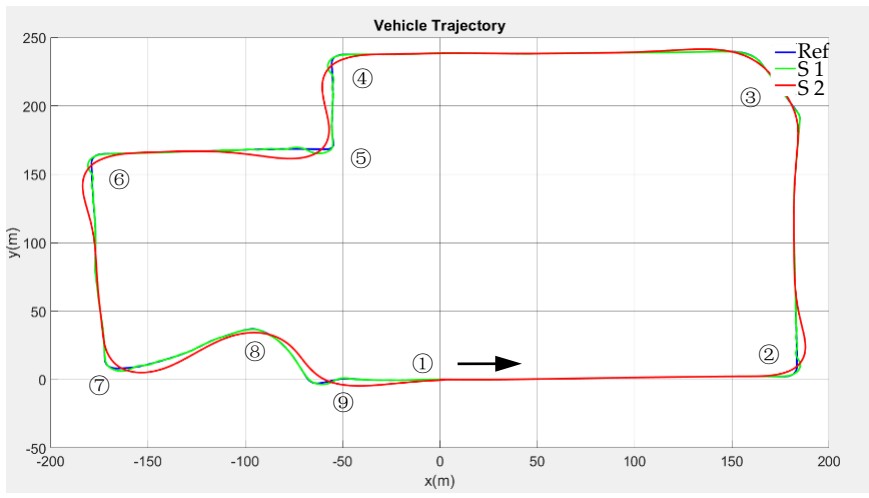

**Figure 11.** Tracking performance on the prescribed route with different emphasis on safety and comfort. Setting 1 has better tracking performance than Setting 2.

### 5.2. Double Lane Change (DLC)

The DLC experiment investigated the driving performance of the proposed MPC strategy and compared it to the traditional proportional–integral–derivative (PID) controller. Both the driver's model and the ISO 3888-2 DLC test scenario were built in in the PreScan simulation platform. Figure 14 shows that the tracking error of the vehicle on the DLC route by the MPC strategy was smaller than that of the traditional PID controller.

Similarly, from Figure 15, the MPC strategy resulted in a better DLC performance than the PID controller in terms of vehicle lateral acceleration.

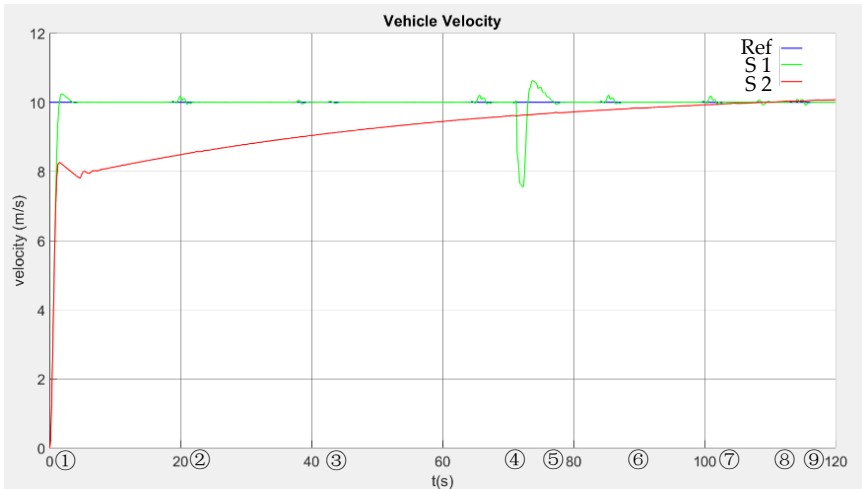

**Figure 12.** Vehicle speed performance on the prescribed route with different emphasis on safety and comfort. Setting 1 follows the reference speed better than Setting 2.

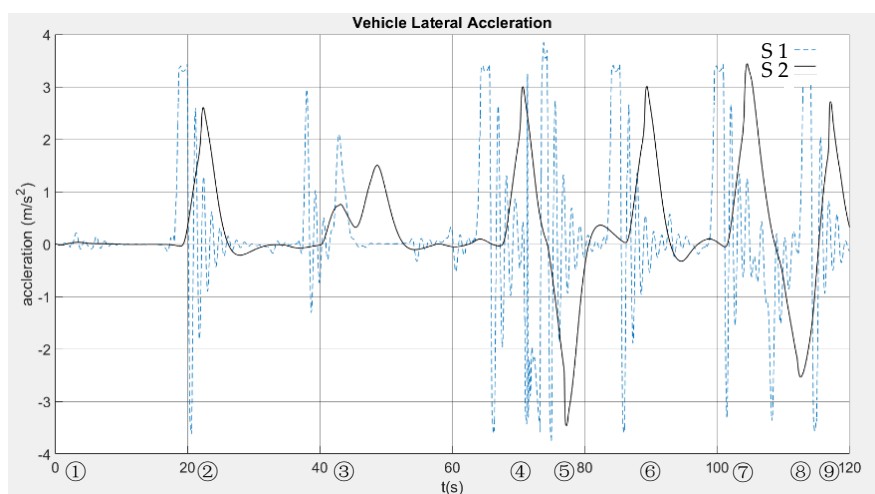

**Figure 13.** Vehicle lateral acceleration curve on the prescribed route with different emphasis on safety and comfort. Setting 2 provides more comfort driving than Setting 1.

### 5.3. Energy Efficiency Simulation

Energy saving is often considered a secondary requirement for self-driving vehicles. With the same scenario of the planned path in Figure 11, the energy efficiency is investigated with 10 sets of prescribed weighting factors in Table 2. The weightings on the vehicle states were still assigned with a high percentage within 45–55% to emphasize safe driving. In addition, it is noticed that the weightings on $w_v$ and $w_{v\delta}$ were given at the same value, and the resultant weightings on $w_\delta$ and $w_{\Delta\delta}$ were assigned within a narrow range between 20.5 and 24, so that the agility and driving comfort were equally important for all these cases. Thus, the major difference was between the weightings $w_{et}$ and $w_{e\varphi}$ on the tracking and heading errors and the weightings $w_p$ and $w_{\dot{p}}$ on the energy saving. The range of the resultant values of $w_p$ and $w_{\dot{p}}$ was between 23 and 34.5.

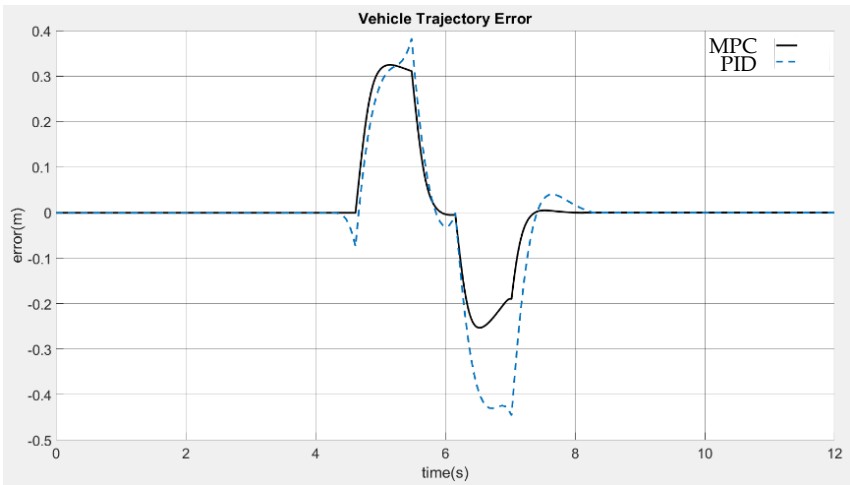

**Figure 14.** Vehicle tracking error on the DLC test by MPC and PID controllers.

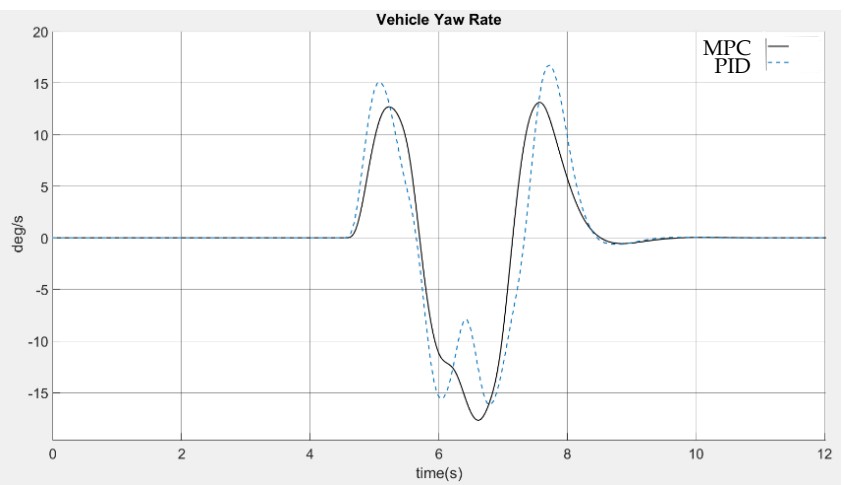

**Figure 15.** Vehicle yaw rate on the DLC test by MPC and PID controllers.

**Table 2.** Weighting factors and energy efficiency improvement.

| | | **Weighting Factors (%)** | | | | | | | | | |
|---|---|---|---|---|---|---|---|---|---|---|---|
| **Case** | | **1** | **2** | **3** | **4** | **5** | **6** | **7** | **8** | **9** | **10** |
| Vehicle states | $w_{et}$ | 12.5 | 17.5 | 15.5 | 12.5 | 16 | 18 | 17.5 | 14 | 15 | 17.5 |
| | $w_{e\varphi}$ | 12.5 | 15.5 | 13 | 12.5 | 15 | 17 | 15.5 | 13 | 13 | 16 |
| | $w_v$ | 10 | 10 | 10 | 10 | 10 | 10 | 10 | 10 | 10 | 10 |
| | $w_{v\delta}$ | 10 | 10 | 10 | 10 | 10 | 10 | 10 | 10 | 10 | 10 |
| Control variables | $w_\delta$ | 8 | 8 | 8 | 10 | 8.5 | 8.5 | 9 | 10 | 9 | 8 |
| | $w_{\Delta\delta}$ | 12.5 | 15 | 15 | 14 | 15 | 15 | 15 | 14 | 14 | 15 |
| | | | | | Weightings on energy saving | | | | | | | |
| | $w_p$ | 14.5 | 12 | 18.5 | 19 | 9 | 18 | 13 | 21 | 14 | 10 |
| | $w_{\dot{p}}$ | 20 | 12 | 10 | 14 | 16.5 | 9 | 10 | 10 | 15 | 13.5 |
| | $w_p + w_{\dot{p}}$ | 34.5 | 24 | 28.5 | 33 | 25.5 | 27 | 23 | 31 | 29 | 23.5 |
| $E_{MPC}$ (QWh) | | 8.28 | 8.68 | 8.45 | 8.31 | 8.64 | 8.60 | 8.49 | 8.32 | 8.39 | 8.73 |
| $\Delta\eta$ (%) | | 7.5 | 3.3 | 5.6 | 7.3 | 3.6 | 4.1 | 2.5 | 7.2 | 6.3 | 2.6 |

Figure 16 shows the time histories of the accelerator pedal angle in percentage (%) for the traditional PID controller and for the three selected cases (1, 3, and 10) to investigate energy efficiency by the proposed MPC strategy for self-driving vehiclecontrol. Since the sampling frequency was 10 Hz in the simulation, the energy consumption was estimated by summing 1200 data of pedal angles, each of which corresponds to an equivalent amount of power. Here, the equivalent power per 1% of pedal angle is assumed to be $Q$ in W/% the total energy consumption can be calculated by

$$E = \frac{Q}{3600} \sum_{k=1}^{1200} p(k) \quad \text{(Wh)}. \tag{32}$$

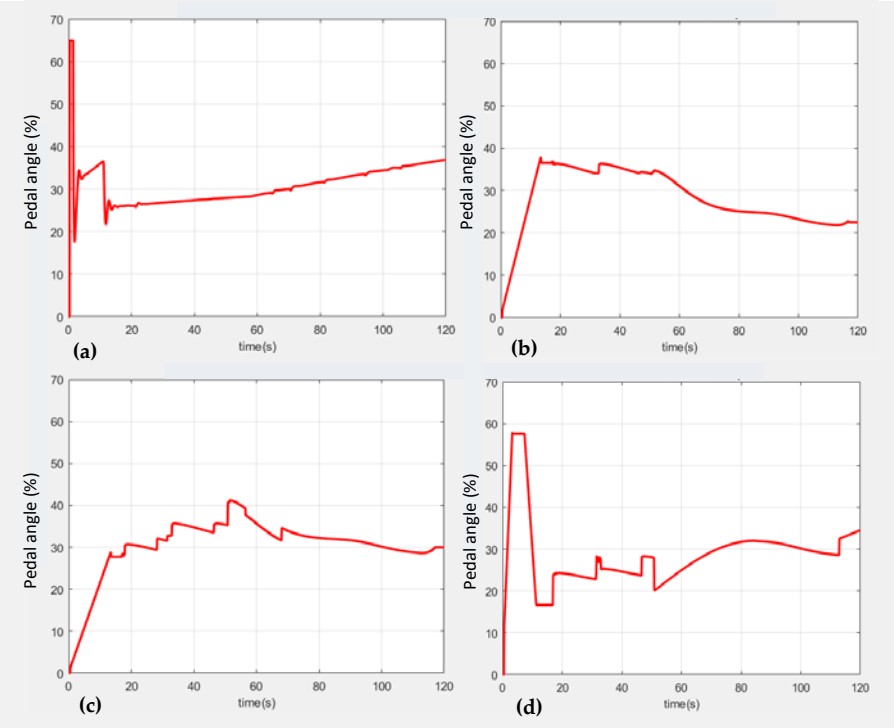

**Figure 16.** Accelerator pedal angle (%) history of (**a**) PID controller, and MPC with (**b**) Case 1, (**c**) Case 3, and (**d**) Case 10 for energy saving simulation. (The area under the curve represents energy consumption).

It is easy to understand that the summation is the estimate of energy consumption that is simply the enclosed area under the curve $p(t)$. The energy efficiency improvement was then calculated by

$$\Delta \eta = \frac{E_{PID} - E_{MPC}}{E_{PID}} \times 100\%, \tag{33}$$

where $E_{PID}$ was 8.96Q (Wh). The $E_{MPC}$ and $\Delta \eta$ of each case are shown in Table 2. The relationship bewteen the energy efficiency improvement and the weighting ($w_p + w_{\dot{p}}$) on energy saving is given in Figure 17. As was predicted, the energy efficiency was better if the weighting on energy saving was higher.

### 5.4. Integral Test of Path Planning and Auto-Parking

In this experiment, we tested manual driving mode, self-driving mode, dynamic path planning, obstacle avoidance, and automatic parking. The parking lot map (Figure 10) was created in the PreScan simulation platform. In Figure 18, a driver on the HiL platform drove the "virtual" vehicle manually to the entrance to the parking lot in about 100 s. After selecting a parking space on the graphical user interface in ROS (RViz), the driver switched to self-driving mode. The vehicle automatically followed the prescribed reference path



without any obstacle. Around 200 s, the vehicle detected an obstacle via RADAR and made an emergency stop. At this point, the hybrid A-star recalculated and updated a new path to avoid the obstacle. Finally, the vehicle passed the obstacle and parked in the selected space. The corresponding curves of the steering angle, yaw rate, and brake pedal signal during the path planning and auto-parking scenarios are shown, respectively, in Figures 19–21, where ordering numbers are provided in correspondence with those in Figure 18.

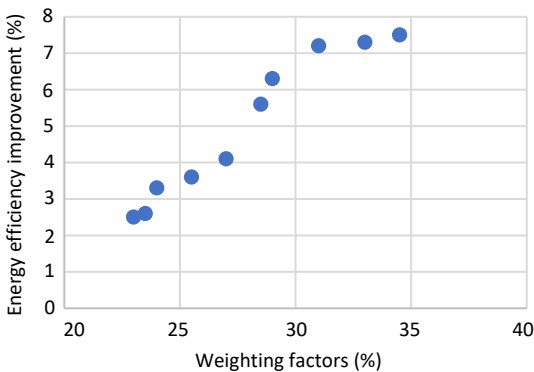

**Figure 17.** Energy efficiency improvement versus the resultant weighing factors ($w_p + w_{\dot{p}}$).

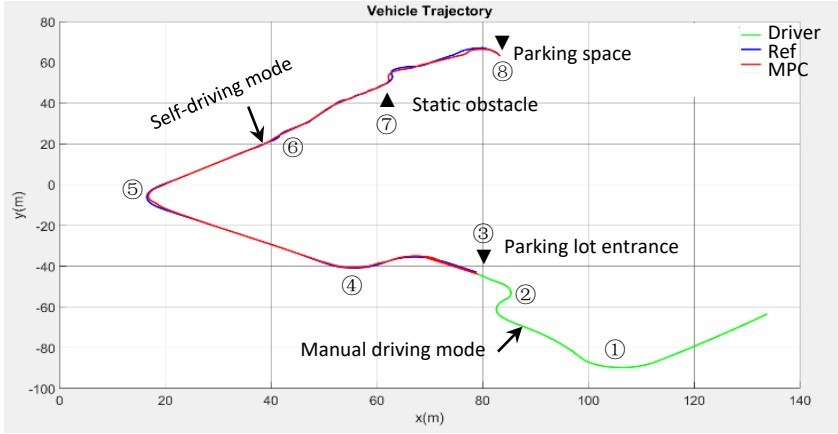

**Figure 18.** 2D map of path planning and auto-parking scenario.

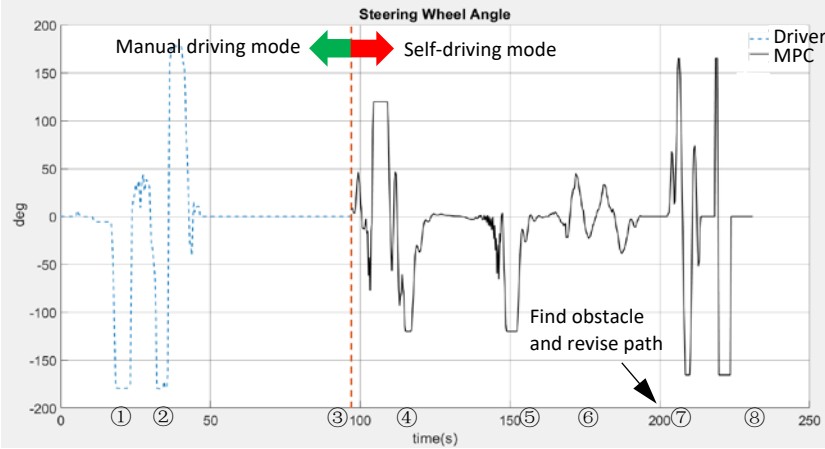

**Figure 19.** Vehicle steering angle curve during the path planning and auto-parking scenario.

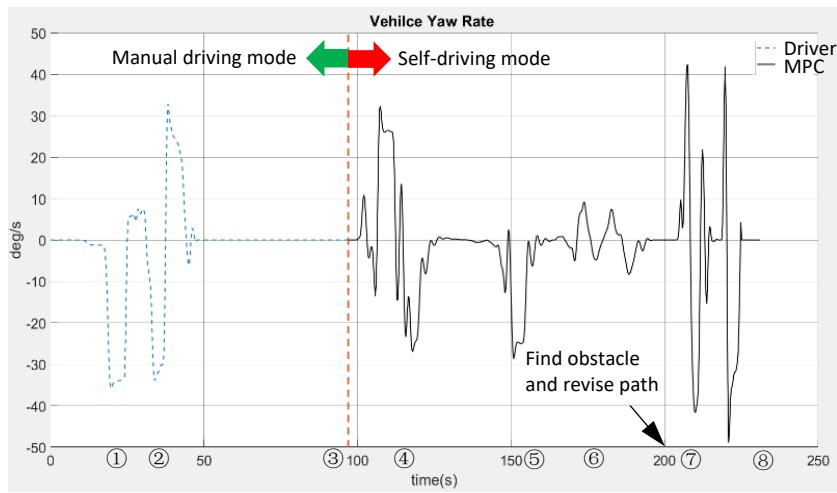

**Figure 20.** Vehicle yaw rate curve during the path planning and auto-parking scenario.

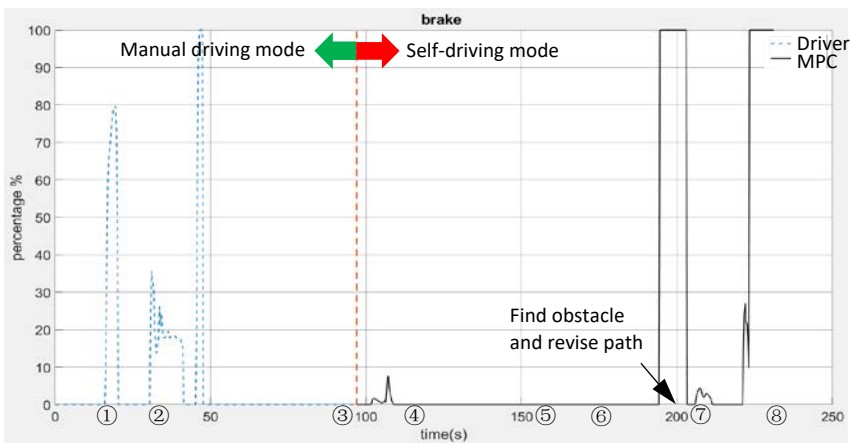

**Figure 21.** Vehicle brake pedal curve during the path planning and auto-parking scenario.

## 6. Conclusions

This paper applied a hybrid A-star algorithm for path planning and the MPC strategy for a self-driving e-van on the HiL simulation platform. The hardware of the platform consisted of an electric power steering system, accelerator, and brake pedals, and an Nvidia drive PX2 on ROS. The vehicle dynamics model, sensors, controller, and test field map were virtually built with the PreScan simulation platform. In comparison with the traditional PID control, the path tracking performance was improved by the proposed MPC strategy by minimizing the prescribed cost function which accounted for vehicle safety, energy-saving, comfort, and agility. Three self-driving scenarios were presented. Self-driving performance was first investigated by two different sets of weighting factors for the cost function of MPC, each accounting for different levels of importance on vehicle safety, energy-saving, comfort, and agility. Second, the stability and tracking performance of the ISO 38888-2 DLC test by the proposed MPC was verified to be better than that by the traditional PID controller. Finally, an integral test was executed on the HiL platform to demonstrate the feasibility of the shift from manual to self-driving mode, obstacle avoidance, dynamic path planning, and auto-parking. This HiL platform can be used for future development of path planning, sensing, perception, decision-making, and control for any kind of self-driving vehicles.

**Author Contributions:** Y.C. and Y.-P.Y. devised and planned the experiments; Y.C. executed the experiments and explained the data; both Y.-P.Y. and Y.C. composed and edited the paper. Both authors have read and agreed to the published version of the manuscript.

**Funding:** This research was financially supported by the Ministry of Science and Technology (MOST) of Taiwan, Republic of China under contract MOST 109-2221-E-002-153.

**Data Availability Statement:** This paper used the open source software package Ipopt for large-scale nonlinear optimization from https://coin-or.github.io/Ipopt/ (accessed on 3 September 2021).

**Acknowledgments:** The authors acknowledge all the engineering support with experimental equipment and HiL simulation platform, from the intelligent mobility division of Mechanical and Mechatronics Systems Research Laboratories, Industrial Technology Research Institute, Hsinchu 31057, Taiwan, China.

**Conflicts of Interest:** The authors claim no conflict of interest.

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
