# Peer review of "Hardware-in-the-Loop Simulation of Self-Driving Electric Vehicles by Dynamic Path Planning and Model Predictive Control"

_electronics, doi:10.3390/electronics10192447_

Round 1

Reviewer 1 Report

The authors are presenting a hardware-accelerated technique for implementing and evaluating advanced control techniques based on Dynamic Path Planning and Model Predictive Control, in order to provide efficient operation of an electric bus model. The article is well-written and structured but more details should be given for better circulating authors'  ideas before potential publication.

For this reason:

The term “hardware-in-the-loop” should be explained more efficiently in the introduction section so as the non-expert reader to better understand the innovation functions introduced by this work and which parts are hardware implemented or software simulated.

In the introduction section, in order the authors to better highlight the need for extending, combining and evaluating control techniques beyond the simple PID one, they may find beneficial the research work addressing similar issues and described in:

Åström, K. J., & Hägglund, T. Advanced PID control. Research Triangle Park, 2006. NC: ISA-The Instrumentation, Systems, and Automation Society.

Vougioukas, S. G. Reactive Trajectory Tracking for Mobile Robots based on Non Linear Model Predictive Control, Proceedings 2007 IEEE International Conference on Robotics and Automation, 2007, pp. 3074-3079. Doi: 10.1109/ROBOT.2007.363939.

Loukatos, D., Petrongonas, E., Manes, K., Kyrtopoulos, I.-V., Dimou, V., & Arvanitis, K.G. (2021). A Synergy of Innovative Technologies towards Implementing an Autonomous DIY Electric Vehicle for Harvester-Assisting Purposes. MDPI Machines, 9(4), 82. Doi: 10.3390/machines9040082

Provide greater details in Section 3, referring to the Hybrid A-Star algorithm.

It would be nice to further clarify if the proposed techniques are applied on the linearized or on the non-linear version of the system under study (the 1st option probably, although the 2nd is more challenging).

An interoperation diagram depicting the separate modules of the control system being implemented (i.e., the PX2 unit, the ROS protocol component, the driving equipment, the operator’s screen, etc) should be welcome, as well as some photos of the assembled simulation components setup.    

The text referring to the curves in the evaluation Figures should be in larger fonts.

In Section 5, more detailed description is needed for the results of each figure and reordering the traces (Section 5.3) so as to provide equal and thus comparable time for both manual and auto vehicle’s operation. The claimed energy efficiency improvements should also be justified.

Author Response

Attached pleased find my file of reply to reviewers' comments.

Reviewer 2 Report

This paper applied a hybrid A-star algorithm for path planning and the MPC strategy for a self-driving e-van on the HiL simulation platform. The study was well presented. It may be improved from the following points:

  • It seems that the new point of the paper was combining hybrid A-star path planning and MPC with vehicle dynamic models for an electric van to simulate self-driving and parking. If so, please provide more references or descriptions that why the combination of A-star path planning and MPC is better than other methods, except PID. Is there any other previous study focused on the same problem while using other methods?
  • “two sets of weighting factors are assigned, as shown in Table I. Setting 1 puts more importance on safety than Setting 2, in terms of 55% versus 35% on the weighting factors ??? and ???. Setting 2 puts more importance on comfort than Setting 1, in terms of 32.5% versus 15% on ??, ?â–³?, and ???.” Please provide more details about Table I. Where is ??? in Table I. How should we understand “55% versus 35% on the weighting factors ??? and ???”. Please kindly explain that how did you choose the values for all these parameters, or the paper will seem like a case study (cases 1 and 2).

Author Response

Attached please find my file of reply to reviewers' comments

Round 2

Reviewer 1 Report

The authors have updated their contribution according to most of the reviewers’ suggestions thus delivering a much more mature manuscript version. Some issues need further elaboration though, prior potential publication.

More specifically, the authors are encouraged to provide more detailed information referring to the grid formation/dimensions points used by the A-star algorithm so as to keep a moderate complexity and also some indicative performance metrics, like CPU utilization and/or completion times for the computational parts of their method, for the given hardware being used.   

Reviewer 2 Report

All of my questions were cleared up
